

# Changes in understory species occurrence of a secondary broadleaved forest after mass mortality of oak trees under deer foraging pressure

Hiroki Itô

Hokkaido Research Center, Forestry and Forest Products Research Institute, Sapporo, Hokkaido, Japan

## ABSTRACT

The epidemic of mass mortality of oak trees by Japanese oak wilt has affected secondary deciduous broadleaved forests that have been used as coppices in Japan. The dieback of oak trees formed gaps in the crown that would be expected to enhance the regeneration of shade-intolerant pioneer species. However, foraging by sika deer *Cervus nippon* has also affected forest vegetation, and the compound effects of both on forest regeneration should be considered when they simultaneously occur. A field study was conducted in Kyôto City, Japan to investigate how these compound effects affected the vegetation of the understory layer of these forests. The presence/absence of seedlings and saplings was observed for 200 quadrats sized 5 m ×5 m for each species in 1992, before the mass mortality and deer encroachment, and in 2014 after these effects. A hierarchical Bayesian model was constructed to explain the occurrence, survival, and colonization of each species with their responses to the gaps that were created, expanded, or affected by the mass mortality of *Quercus serrata* trees. The species that occurred most frequently in 1992, *Eurya japonica*, *Quercus glauca*, and *Cleyera japonica*, also had the highest survival probabilities. Deer-unpalatable species such as *Symplocos prunifolia* and *Triadica sebifera* had higher colonization rates in the gaps, while the deer-palatable species *Aucuba japonica* had the smallest survival probability. The gaps thus promoted the colonization of deer-unpalatable plant species such as *Symplocos prunifolia* and *Triadica sebifera*. In the future, such deer-unpalatable species may dominate gaps that were created, expanded, or affected by the mass mortality of oak trees.

# INTRODUCTION

Many coppices have been abandoned for socio-economic reasons, such as the replacement of woody fuels with fossil fuels in Europe (*Rackham, 2008*; *Müllerová, Hédl & Szabó, 2015*; *Svátek & Matula, 2015*). This abandonment parallels that of Japan (*Suzuki, 2013*), and a considerable number of deciduous oak forests grown from such coppices have suffered from the mass mortality of oak trees (*Kuroda, Osumi & Oku, 2012*; *Nakajima & Ishida, 2014*) caused by Japanese oak wilt (*Kuroda, Osumi & Oku, 2012*). The direct cause of the mortality is a pathogenic species of fungus *Raffaelea quercivora* Kubono et Shin. Ito,

Corresponding author
Hiroki Itô, abies.firma@gmail.com

which is carried by the ambrosia beetle *Platypus quercivorus* Murayama (*Kubono & Ito, 2002*; *Kinuura & Kobayashi, 2006*). In addition, it has been pointed out that the coppice abandonment is an indirect cause of the epidemic because the ambrosia beetles reproduce more successfully in large oak stems, which are more abundant as abandoned oak stems grow up (*Kobayashi & Ueda, 2005*). There is another possible reason why the abandonment indirectly affected the epidemic: in the period when coppices were managed, oak stems that died from the wilt were quickly felled and utilized for fuel or charcoal, and as a result, it prevented outbreaks of the disease (*Ida & Takahashi, 2010*).

The mass mortality altered the structures of damaged oak forests. The two major deciduous oak species in Japan, *Quercus crispula* Blume and *Quercus serrata* Murray, are vulnerable to the pathogenic fungus. *Nakajima & Ishida (2014)* showed that 80 ± 19% (mean ± standard deviation) stems of *Quercus crispula* died, while 34 ± 19% stems of *Quercus serrata* died. *Naka (1982)* studied an old-growth evergreen forest in the Kasugayama Forest Reserve in Nara City, which is located about 40 km south of the study site, and showed that the major gap generator was typhoons, and that the interval between the typhoon disturbances was 6.57 years with a tree fall rate for overstory trees of 0.84 trees/ha/year, and that the canopy opening rate was 55.6 m$^2$/ha/year. Though these rates are not directly applicable to deciduous secondary oak forests, the gaps created by the mortality would not be negligible for oak-dominated forests. If *Quercus serrata* covers 30% of the canopy and 30% of them die, 900 m$^2$/ha of the canopy will become gaps. How such damaged oak forests regenerate depends on the circumstances; sub-canopy trees might grow to canopy trees in some cases (*Itô, Igarashi & Kinuura, 2009*), and dense floor vegetation such as dwarf bamboo might inhibit regeneration in other cases (*Itô, Kinuura & Oku, 2011*; *Saito & Shibata, 2012*). In the latter case, the damaged forests may lack a canopy layer for a long time.

For the last several decades, herbivory by overabundant deer populations has negatively affected forest vegetation in Europe and North America (*Rooney, 2001*; *Côté et al., 2004*; *Rackham, 2008*). This is paralleled in Japan: sika deer (*Cervus nippon* Temminck) is one major inhibitor of forest regeneration (*Takatsuki, 2009*; *Suzuki, 2013*; *Iijima & Nagaike, 2015*). However, little is known about how regeneration proceeds after mass mortality under deer foraging pressure (*Obora, Watanabe & Yokoi, 2013*). Gap formation should improve light conditions on the forest floor and promote the regeneration of many tree species (*Suzuki, 2013*). However, excessive browsing should inhibit regeneration, except for unpalatable and browse-tolerant species (*Shimoda et al., 1994*). Therefore, a combination of both factors might promote the regeneration of species that are unpalatable and/or browse-tolerant to deer. This in turn may alter the species composition of forests that regenerate from the mass mortality of oak trees over time.

It is important for the management of secondary forests to predict what tree or shrub species will be recruited in forest stands damaged by the mass mortality of oak trees and deer impact. In a previous study, *Itô (2015)* described the changes in the canopy/sub-canopy and understory layers of a damaged forest by comparing vegetation before and after mass mortality of oak trees and deer foraging. It was found that regenerating species were limited to the originally abundant species, such as *Quercus glauca* Thunb.,

*Cleyera japonica* Thunb., and *Eurya japonica* Thunb. var. *japonica*, as well as to species unpalatable to deer, such as *Symplocos prunifolia* Siebold et Zucc. and *Triadica sebifera* (L.) Small. However, the study only described the changes in species occurrence and failed to estimate the specific probabilities of survival and colonization. In this study, the previous data from the understory layers were reanalyzed using a hierarchical Bayesian model that explicitly incorporates probabilities of ecological processes such as occurrence, survival, and colonization. A hierarchical model incorporating random species effects also makes it possible to estimate those probabilities by species, "borrowing strength from the ensemble" (*Kéry & Schaub, 2012*). These advantages should be of use in predicting which species will dominate such stands in the future.

## MATERIALS AND METHODS

### Study site

The field data were collected in the Ginkakuzi-san (also spelled Ginkakuji-san) National Forest located in Kyôto City, Japan (35.029°N, 135.801°E). The yearly average temperature from 1981 to 2010 was 15.9 °C and the average precipitation was 1491.3 mm at the Kyôto Local Meteorological Office. Elevation of the forest was 100–290 m above sea level, and the forest was in the warm temperate zone. The national forest was protected for its landscape and erosion control, and most of it was situated in the buffer zone of the UNESCO world heritage site, historic monuments of ancient Kyôto. The fieldwork was conducted with permission under an agreement between the Kyôto-Ôsaka District Forest Office and the Forestry and Forest Products Research Institute.

In the 1930s, most of the forest canopy consisted of a mix of pines (*Pinus densiflora* Siebold et Zucc.) and broadleaved trees including oak (*Quercus serrata*). After the 1960s, many pine trees had died due to the pine wilt disease. Recently, most of the national forest has been covered with a secondary broadleaved forest consisting of many species such as evergreen oak *Quercus glauca*, evergreen subcanopy species *Symplocos prunifolia*, and deciduous tree species *Ilex macropoda* Miq., although conifers (*Cryptomeria japonica* (L.f.) D. Don and *Chamaecyparis obtusa* (Siebold et Zucc.) Endl.) were planted in a small part of the area (*Itô, 2007*). In 2005, the mass mortality of oaks was first recognized in the eastern part of Kyôto City, in which the Ginkakuzi-san National Forest is located, and then the damage expanded (*Itô, 2015*). In addition, damage by sika deer resulting from browsing and bark-stripping has been noticeable over the same period. Deer had been seldom seen in the 1990s (pers. obs.), but by this time inhabited the forest throughout the year (*Itô, 2015*).

In 1992, a 0.5 ha (100 × 50 m) plot was established on a south-facing slope in the national forest at an elevation of 140–195 m. The average slope inclination was about 30° and the surface geology was granite. All the stems in the plot were marked and their diameters at breast height (dbh) were measured in 1993, 1996, 1999, 2002, 2005, and 2014. Mainly due to the mass mortality of oak trees *Quercus serrata*, the basal area in the plot decreased from 43.3 $m^2$/ha in 2005 to 39.5 $m^2$/ha in 2014, while the number of stems in the plot increased from 1,554 to 1,645 (3,108 stems/ha to 3,290 stems/ha in stem density). There were 36 *Quercus serrata* stems in the plot in 2005, and 21 of them died by 2014.

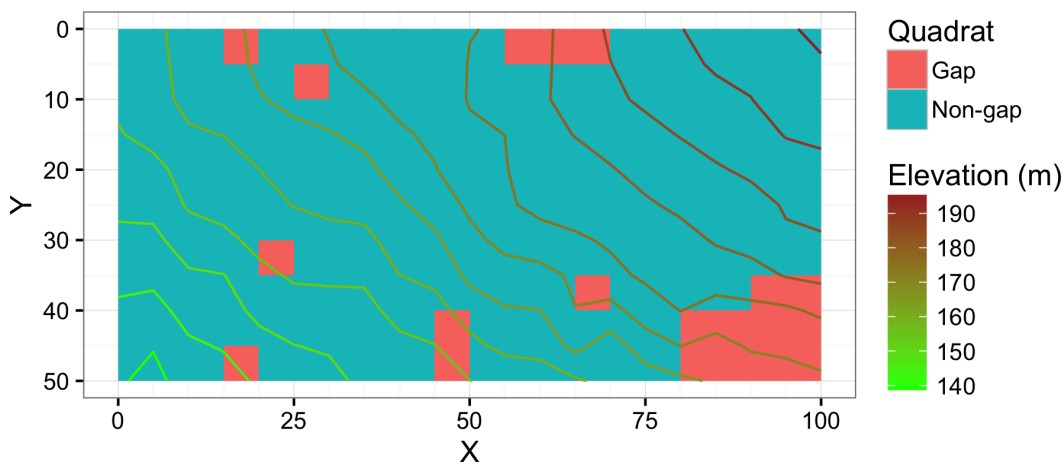

**Figure 1** **Map showing the study site.** Gap denotes the gap quadrats created, expanded, or affected by the mass mortality of oak trees. Non-gap denotes the rest of the quadrats. Units of the X and Y axes are in meters.

These trees were most likely killed by Japanese oak wilt. This affected the forest structure in the plot via the formation of new canopy gaps or by expanding existing gaps. In some area, the mortality indirectly affected the forest floor via treatments of dead stems by the forest office (cutting down the stems, cutting the fallen stems into pieces, and disinfecting them) (*Itô, 2015*). On the other hand, none of the evergreen oaks *Quercus glauca* died from the disease, though some were attacked by the ambrosia beetles, being less vulnerable than *Quercus serrata* (*Murata, Yamada & Ito, 2005*; *Murata et al., 2009*). In the understory layer, *Quercus glauca* and some evergreen shrub species such as *Eurya japonica* and *Cleyera japonica* were frequently observed. Overall changes in the species composition of the site from 1990s to 2010s were reported in *Itô (2015)*.

## Data collection

The plot was divided into 200 quadrats sized 5 m × 5 m. In 1992, all woody plant seedling or saplings (age ≥ 1 year and dbh < 3 cm) were tallied by species. The plots were resampled in 2014. In 2014, each quadrat was classified into inside or outside of gaps formed by the death of oak trees or older gaps that were affected in addition by fallen dead oak trees. Quadrats on the boundary were visually classified; if an open area which was created, expanded, or affected by the mass morality occupied most of the quadrat, the quadrat was classified as a gap quadrat. Twenty of 200 quadrats were classified as gaps created or affected by oak death (Fig. 1). The size of the largest gap was approximately 250 m² consisting of 10 adjacent quadrats.

In 1992, 55 species were observed and 58 species were observed in 2014 (*Itô, 2015*), amounting to 75 species in total over both years. The species with highest frequency of occurrence over quadrats in 1992 were *Quercus glauca* and *Eurya japonica*, both of which were observed in 184 of 200 quadrats. The mean and median numbers of quadrats in 1992 in which a given species occurred were 19.5 and 4, respectively, excluding species which were not present in 1992. On the other hand, the maximum in 2014 was 193 for *Quercus*

*glauca*. The mean and median were 15.6 and 5, respectively, excluding species which did not occur in 2014 in any quadrats.

Data analysis was conducted for 42 of the species that were observed in more than 5 quadrats over both sample years (200 quadrats × 2 observations).

## Statistical modeling

A hierarchical Bayesian model was constructed to determine how species were affected by the recent environmental changes in the forest.

The presence/absence (presence = 1, absence = 0) of species $i$ in quadrat $j$ in the year 1992 ($y_{1ij}$) and 2014 ($y_{2ij}$) was assumed to follow the Bernoulli distribution given the occurrence probability $\psi_{1ij}$ and $\psi_{2ij}$, as follows,

$$y_{1ij} \sim \text{Bernoulli}(\psi_{1ij})$$
$$y_{2ij} \sim \text{Bernoulli}(\psi_{2ij}).$$

To be exact, the "presence/absence" was "detection/nondetection" (*Dorazio et al., 2006*; *Kéry & Schaub, 2012*). It has been pointed out that detection probability should be considered to correctly estimate population properties such as the occurrence rate or survival rate, even if the observed objects are plants (*Kéry, 2004*; *Chen et al., 2009*; *Chen et al., 2013*). The present study had only one observation for each survey year. However, the quadrat size was rather small (5 m × 5 m) and the whole of each quadrat was explored, so I expected that the detection probability should be near to one and therefore "detection/non-detection" was regarded as "presence/absence" in this study. *Chen et al. (2009)* showed that the detection probability asymptotically approaches one with larger survey efforts.

The parameter of occurrence in 1992, $\psi_{1ij}$, was formulated as follows:

$$\text{logit}(\psi_{1ij}) = \beta_o + \epsilon_{oi} + r_j,$$

where $\beta_o$ denotes an intercept or overall mean of $\psi_1$ on the logit scale, and $\epsilon_{oi}$ denotes the random species effect on the intercept of species $i$. The parameter $r_j$ denotes a spatially autocorrelated random effect of quadrat $j$.

The parameter of occurrence in 2014, $\psi_{2ij}$, was formulated as follows:

$$\psi_{2ij} = y_{1ij}\phi_{ij} + (1 - y_{1ij})\gamma_{ij},$$

where parameter $\phi_{ij}$ denotes the 'survival' probability that species $i$ was present in quadrat $j$ in 1992 and still present in 2014. The parameter $\gamma_{ij}$ denotes the 'colonization' probability that species $i$ was absent in quadrat $j$ in 1992 but present in 2014.

The parameters of survival $\phi_{ij}$ and colonization $\gamma_{ij}$ were formulated as follows:

$$\text{logit}(\phi_{ij}) = \beta_s + \epsilon_{si} + (\beta_{sg} + \epsilon_{sgi})g_j$$
$$\text{logit}(\gamma_{ij}) = \beta_c + \epsilon_{ci} + (\beta_{cg} + \epsilon_{cgi})g_j,$$

where the parameters $\beta_s$ and $\beta_c$ are intercepts or overall means of $\phi$ and $\gamma$ on the logit scale, respectively. The parameters $\epsilon_{si}$ and $\epsilon_{ci}$ are random species effects on the intercepts,

$\beta_{sg}$ and $\beta_{cg}$ are coefficients of the gap predictor $g_j$ (0: non-gap quadrats, 1: gap quadrats affected by oak mortality), and $\epsilon_{sgi}$ and $\epsilon_{cgi}$ are random species effects on the coefficients.

Priors of the random species effects were defined hierarchically; hyperparameters, $\sigma_o$, $\sigma_s$, $\sigma_{sg}$, $\sigma_c$, and $\sigma_{cg}$ scaled the distribution of $\epsilon_{oi}$, $\epsilon_{si}$, $\epsilon_{sgi}$, $\epsilon_{ci}$, and $\epsilon_{cgi}$, respectively, as follows:

$$\epsilon_{oi} \sim \text{Normal}(0, \sigma_o^2)\text{T}(-10, 10),$$

where $\text{Normal}(0, \sigma^2)\text{T}(-10, 10)$ denotes a normal distribution truncated at $-10$ and 10; the truncation was incorporated to stabilize the logit scale parameters (*Kéry & Schaub, 2012*). Priors of the parameters $\beta_o$, $\beta_{op}$, $\beta_{os}$, $\beta_s$, $\beta_{sg}$, $\beta_c$, and $\beta_{cg}$ were defined as $\text{Normal}(0, 10^4)\text{T}(-10, 10)$. The prior of the spatial effect $r_j$ was defined as an intrinsic conditional autoregressive model as follows:

$$r_j | r_{-j} \sim \text{Normal}\left(\sum_{k \neq j} \frac{w_{jk} r_j}{w_{j+}}, \frac{\sigma_r^2}{w_{j+}}\right),$$

where $r_{-j}$ denote the values of $r$ except for the quadrat $j$, a variable $w_{jk}$ was defined to be 1 if quadrat $j$ and quadrat $k$ are adjacent, and 0 if not, and $w_{j+}$ was defined to be $\sum_k w_{jk}$. The parameter $\sigma_r^2$ denotes a variance of the random effect.

Presence/absence data for 42 species, which were observed in more than 5 quadrats in total over two surveys, 1992 and 2014, were used for the parameter estimation. To estimate the posterior distribution for each parameter, the Markov chain Monte Carlo (MCMC) method was adopted; this simulation method generates Markov chains drawing values from the target posterior distributions if the chains converge to stationary distributions (*Gelman et al., 2013*). Four parallel chains were generated in this study, and each of them had 13,000 iterations while the first 3,000 iterations were dropped as burn-in. The MCMC sample was taken from the three chains with 10 thinning intervals, so that the sample size was 4,000. OpenBUGS 3.2.3 (*Lunn et al., 2009*) was used for the computation. The BUGS code is available in Supplemental Information 2. To check the convergence, Gelman–Rubin statistics ($\hat{R}$) were calculated (*Gelman & Rubin, 1992*; *Brooks & Gelman, 1998*; *Gelman et al., 2013*). When the Markov chains successfully converge, the value of $\hat{R}$ becomes nearly one. If the value of $\hat{R}$ is no larger than 1.1, the chains are usually regarded as converged (*Kéry & Schaub, 2012*).

## RESULTS

For each parameter, the values of $\hat{R}$ were no larger than 1.1, so that the Markov chains seemed to reach convergence. However, some random species effects on coefficients of gaps had rather wide posteriors as mentioned later. Those parameters might lack enough information to obtain satisfactory estimates.

Posterior mean, median, and 95% credible interval (CI) of the overall occurrence probability in 1992 (probability that a species was present in a quadrat in 1992), $\beta_o$, was estimated to be $-3.25$ for the posterior mean, $-3.25$ for the median, and $-3.99$ to $-2.54$ for the 95% CI (Table 1). The value $-3.25$ on the logit scale is equivalent to 0.037
_________________________________________________

**Table 1  Parameter estimates (posterior mean, standard deviation (SD), and 2.5%, 5%, 50%, 95%, and 97.5% quantiles) other than random effects.** Positive values of the intercepts ($\beta_o$, $\beta_s$, and $\beta_c$) on the logit scale mean that the corresponding probabilities are larger than 0.5, and vice versa. Positive values of the coefficients of gap ($\beta_{sg}$ and $\beta_{cg}$) mean that the gap increases the corresponding probabilities. Parameters $\sigma_o$ and below are standard deviations of the corresponding predictors.

|  | Mean | SD | 2.5% | 5% | 50% | 95% | 97.5% |
|---|---|---|---|---|---|---|---|
| $\beta_o$ | −3.25 | 0.37 | −3.99 | −3.88 | −3.25 | −2.65 | −2.54 |
| $\beta_s$ | −2.60 | 0.67 | −4.11 | −3.81 | −2.57 | −1.61 | −1.44 |
| $\beta_{sg}$ | −0.17 | 0.65 | −1.54 | −1.27 | −0.14 | 0.81 | 1.04 |
| $\beta_c$ | −3.81 | 0.32 | −4.45 | −4.33 | −3.81 | −3.28 | −3.18 |
| $\beta_{cg}$ | 1.51 | 0.31 | 0.86 | 0.98 | 1.53 | 1.99 | 2.08 |
| $\sigma_o$ | 2.27 | 0.30 | 1.76 | 1.82 | 2.24 | 2.81 | 2.95 |
| $\sigma_s$ | 2.89 | 0.57 | 1.97 | 2.07 | 2.84 | 3.91 | 4.13 |
| $\sigma_{sg}$ | 1.51 | 0.84 | 0.19 | 0.31 | 1.40 | 3.07 | 3.48 |
| $\sigma_c$ | 1.89 | 0.28 | 1.43 | 1.48 | 1.86 | 2.40 | 2.51 |
| $\sigma_{cg}$ | 1.35 | 0.32 | 0.81 | 0.88 | 1.32 | 1.93 | 2.07 |
| $\sigma_r$ | 0.44 | 0.10 | 0.27 | 0.29 | 0.44 | 0.60 | 0.65 |

$(= 1/(1 + \exp(3.25)))$ on the probability scale. Therefore, a species was expected to occur in 3.7% of quadrats on average in 1992. In the same manner, the 95% CI was 1.8–7.3% on the probability scale.

The overall survival probability (probability that a species was present in a quadrat in 1992 and still present in the same quadrat in 2014), $\beta_s$, was estimated to be −2.60 for the posterior mean, −2.57 for the median, and −4.11 to −1.44 for the 95% CI. The mean value was equivalent to a probability that a species occurring in 1992 surviving in the same quadrat in 2014 was expected to be 6.9%, and the 95% CI was 1.6–19.2%.

On the other hand, the overall colonization probability (probability that a species was absent in a quadrat in 1992 but present in the same quadrat in 2014), $\beta_c$, was estimated to be −3.81 for the posterior mean, −3.81 for the median, and −4.45 to −3.18 for the 95% CI. The value −3.81 was equivalent to 0.022 on the probability scale; so that the probability that a species which was absent in a quadrat in 1992 had colonized into the quadrat in 2014 was expected to be 2.2%, and the 95% CI was 1.2–4.0%.

Posterior mean of coefficients of the gap effect on survival $\beta_{sg}$ and colonization $\beta_{cg}$ were −0.17 and 1.51, respectively, and $\beta_{cg}$ did not include zero in the 95% CI (0.86–2.08), while $\beta_{sg}$ included zero in the 90% CI (−1.27–0.81). The mean value 1.51 of $\beta_{cg}$ meant that gaps increased average colonization probability from 2.2% to 9.1% $(= 1/(1 + \exp(3.81 - 1.51)))$ on average.

Random species effects on occurrence probability in 1992 ($\epsilon_o$) are shown in Fig. 2. *Eurya japonica* and *Quercus glauca* had the largest value, followed by *Cleyera japonica*, *Aucuba japonica* Thunb. var. *japonica*, *Ilex crenata* Thunb., *Photinia glabra* (Thunb.) Maxim., and so on. The posterior mean for *Eurya japonica* and *Quercus glauca* was 5.71, and this meant that the occurrence probability of the species increased from an overall mean of 3.7%–92.1% $(= 1/(1 + \exp(3.25 - 5.71)))$. The 95% CI of the random species effect for *Eurya japonica*, 4.85–6.62, corresponded to 83.2–96.7% for the occurrence probability for the

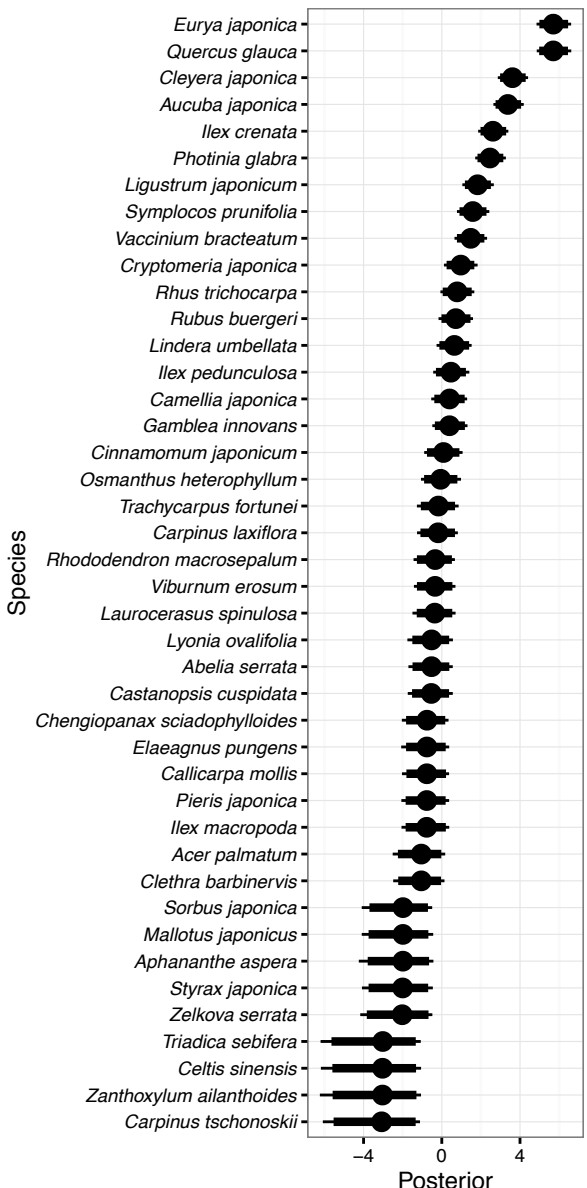

**Figure 2  Random species effects on occurrence in 1992 $\epsilon_o$.** Thin lines denote 95% credible intervals (CI), thick lines denote 90% CI, and circles denote medians. Larger positive values in the posterior indicate that the species will be more likely to be present in 1992 than the overall mean, and negative values indicate that the species will be less likely.

species. In the same manner, the expected occurrence probabilities were 59.1% for *Cleyera japonica*, 53.2% for *Aucuba japonica*, 34.8% for *Ilex crenata*, and 31.6% for *Photinia glabra*. On the other hand, *Carpinus tschonoskii* Maxim., *Celtis sinensis*, *Zanthoxylum ailanthoides* Siebold et Zucc., and *Triadica sebifera* had the smallest values because these species were not detected in 1992 (Fig. 2). The posterior mean for *Carpinus tschonoskii* was −3.21 on the logit scale and the expected occurrence was 0.2% ($= 1/(1 + \exp(3.25 + 3.21))$) on the probability scale. The 95% CI of the species, −6.08 to −1.11, corresponded to an occurrence probability of 0.0–1.3% on the probability scale.

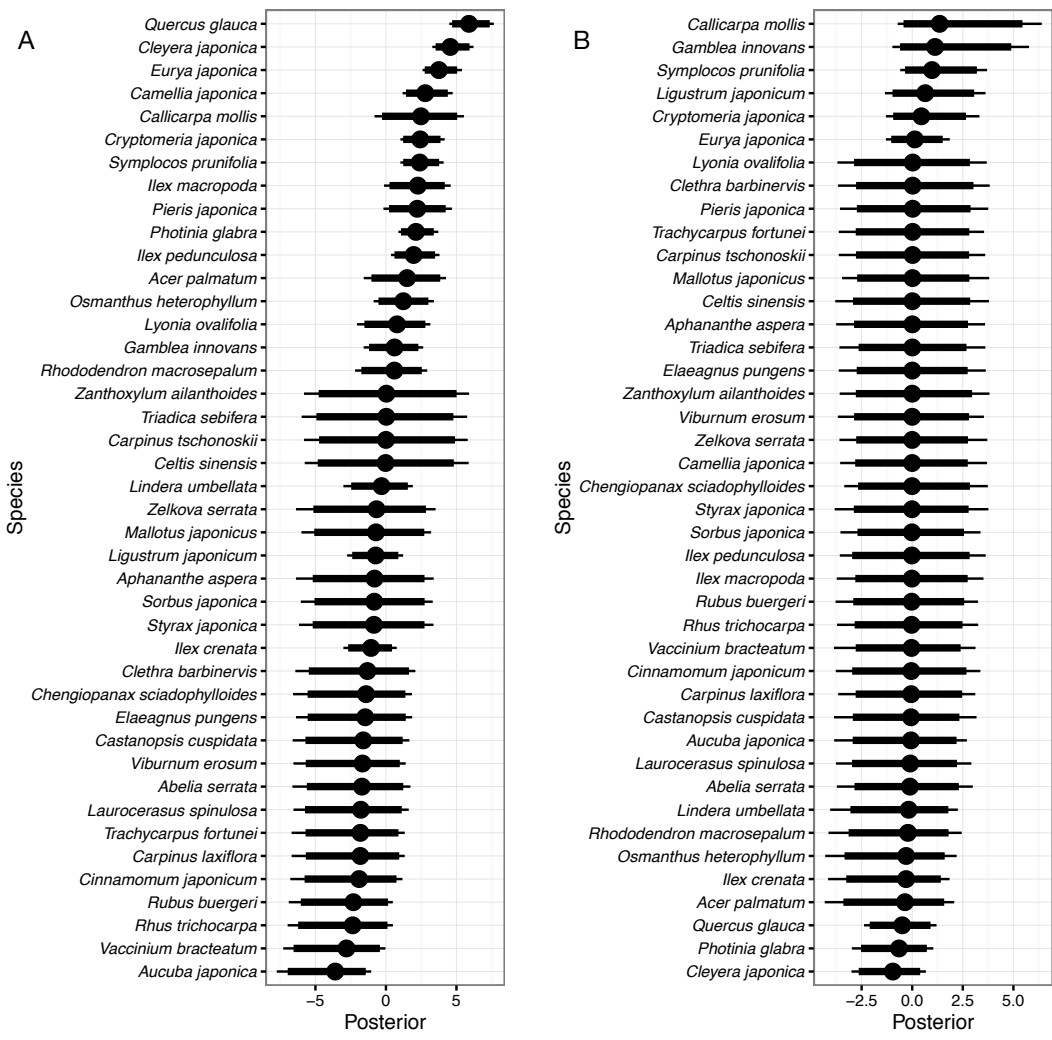

**Figure 3** **Random species effects on survival.** (A) $\epsilon_s$, on the intercept, and (B) $\epsilon_{sg}$, on the coefficient of the effect of gap. Thin lines denote 95% credible intervals (CI), thick lines denote 90% CI, and circles denote medians. Larger positive values in the posterior indicate that a species that had been present in a quadrat in 1992 will be more likely to be present in the same quadrat in 2014 than the overall mean, and negative values indicate that the species will be less likely.

Random species effects on the intercept of survival from 1992 to 2014 ($\epsilon_s$) are shown in Fig. 3A. *Quercus glauca* had the largest value, and *Cleyera japonica*, *Eurya japonica*, and *Camellia japonica* L. followed. The posterior mean of *Quercus glauca* was 5.93, and the expected survival probability was 96.5% ($= 1/(1 + \exp(2.60 - 5.93))$) without the gap effects. The expected survival probabilities were 88.3% for *Cleyera japonica*, 76.7% for *Eurya japonica*, and 55.7% for *Camellia japonica*. On the other hand, the posterior of *Aucuba japonica* was less than zero within 95% CI (Fig. 3A). The posterior mean of the random effect was −3.80 and the expected value of survival probability was 0.2%. The posterior mean of *Vaccinium bracteatum*. Thunb was the second smallest and the value was −3.03; the expected survival probability was 0.4%. Species without surviving individuals had nearly zero means and wider CIs, such as for *Zanthoxylum ailanthoides*. Random

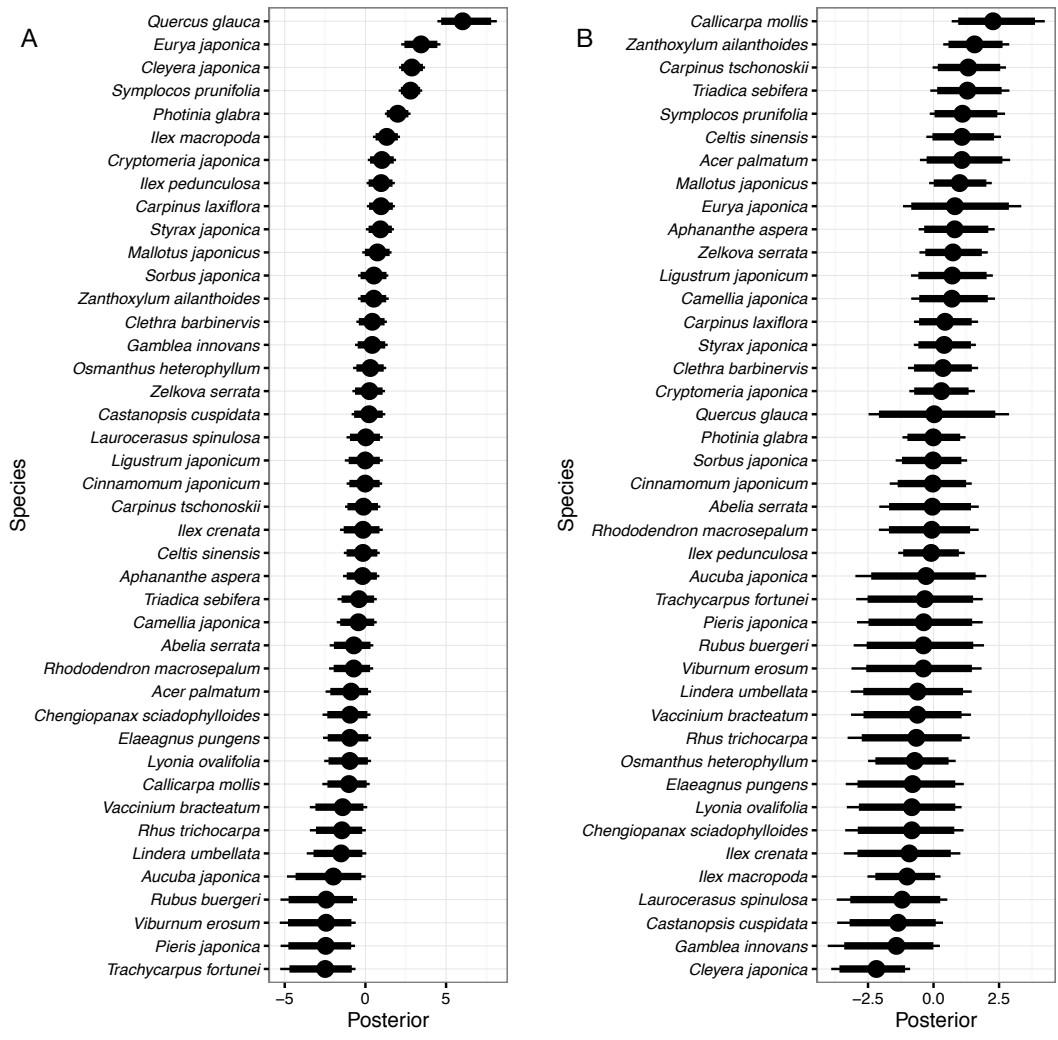

**Figure 4** **Random species effects on colonization.** (A) $\epsilon_c$, for the intercept and (B) $\epsilon_{cg}$, for the coefficient of the gap effect. Thin lines denote 95% credible intervals (CI), thick lines denote 90% CI, and circles denote medians. Larger positive values in the posterior indicate that species that had been absent in a quadrat in 1992 will be more likely to be present in the same quadrat in 2014 than the overall mean, and negative values indicate that the species will be less likely.

species effects on the survival coefficients of the gap predictor were rather small as absolute values and included zero in their 90% CI for all species (Fig. 3B).

Random species effects on intercepts of colonization from 1992 to 2014 were the largest in *Quercus glauca*, followed by *Eurya japonica*, *Cleyera japonica*, *Symplocos prunifolia*, *Photinia glabra*, and so on (Fig. 4A). The posterior mean of the random effect for *Quercus glauca* was 6.09 and the expected colonization probability without the gap effects was 90.7%. In the same manner, the expected colonization probabilities for *Cleyera japonica*, *Symplocos prunifolia*, and *Photinia glabra* were 28.3%, 26.7%, and 14.1%, respectively.

Random species effects on the colonization coefficient of the gap predictor were the largest for *Callicarpa mollis* Siebold et Zucc., followed by *Zanthoxylum ailanthoides*, *Carpinus tschonoskii*, *Triadica sebifera*, and so on. They were the smallest in *Cleyera japonica*.

The posterior mean of the random effect for *Callicarpa mollis* was 2.32, so that the expected colonization probability for the species in the gap quadrats was estimated to be 25.9% $(= 1/(1 + \exp(3.81 - 1.51 + 1.07 - 2.32)))$, as the posterior mean of the random species effect on the colonization for the species was $-1.07$. Some species such as *Quercus glauca* had wider CI in their posteriors. This might be because there was little information for such species due to the small number of colonizers within the gaps (Fig. 4B).

Complete estimates of random effects are available in Table S3.

## DISCUSSION

The mean probability of overall occurrence in 1992 for individual species was estimated to be only 3.7% for each quadrat. This reflects the fact that many species occurred only in a small fraction of the quadrats. The most frequent species in 1992 were all evergreen tree or shrub species such as *Eurya japonica*, *Quercus glauca*, *Cleyera japonica*, *Aucuba japonica*, *Ilex crenata*, and *Photinia glabra*; the expected occurrence probabilities of these species were all larger than 30%. Species with large values of random species effects concerning occurrence corresponded to these species as a matter of course (Fig. 2). On the other hand, the least frequently occurring species were deciduous trees or shrubs such as *Carpinus tschonoskii*, *Zanthoxylum ailanthoides*, *Celtis sinensis* Pers., and *Triadica sebifera*; these species were not detected in the quadrats in 1992. The canopy of the forest was almost closed in 1992, so that shade-tolerant evergreen tree or shrub species dominated the understory layer and deciduous early-successional species infrequently occurred.

Overall survival and colonization probability by individual species were expected to be 6.9% and 2.2%, respectively, on average. These may seem small, but some species have large random effect values allowing them to survive or colonize at high probabilities. The average colonization probability was estimated to increase in the gap quadrats. This was in accord with the preceding study showing that gap creation increased floor species richness even under deer pressure (*Suzuki, 2013*).

The species that had the highest survival probability in the non-gap quadrats was *Quercus glauca* (Fig. 3A); the mean probability of survival was expected to be 96.5%. *Quercus glauca* produces many sprouts (*Cai, 2000*), and the sprouts may survive when the main stems die. This trait might contribute to the high survival probability. *Cleyera japonica*, *Eurya japonica*, and *Camellia japonica* followed *Quercus glauca* in their survival probabilities. These species, including *Quercus glauca*, were all evergreen broadleaved species that could survive under a closed canopy, as shown by their occurrence. In contrast, *Aucuba japonica*, a species of evergreen shrub, had the smallest survival probability at 0.2%. This species was found in 107 of 200 quadrats in 1992, but never found in 2014. *Aucuba japonica* is known to be a preferred food for sika deer and is vulnerable to deer browsing (*Hashimoto & Fujiki, 2014*). The encroachment of deer likely explains its small survival probability. The posterior mean of the random effect was the second smallest in *Vaccinium bracteatum*. There is relatively little information on the palatability of *Vaccinium bracteatum* for sika deer, and some reports refer to it as a food plant, while others refer to it as unpalatable (*Nakajima, 1929*; *Kabaya, 1988*; *Takatsuki, 1989*; *Hashimoto & Fujiki, 2014*). The present
results suggest that the species is vulnerable to deer impacts. The random species effects on the survival coefficients of the gap predictor were not so clear (Fig. 3B). This may due both to the small sample size of the gap quadrats, and to the small number of light-demanding species occurring in 1992.

The colonization probability in the non-gap quadrat was highest in *Quercus glauca*, and evergreen shrub or sub-canopy trees followed (Fig. 4A). Among them, *Quercus glauca* had an especially high colonization probability; the expected value was 90.7%. This might be because numerous seeds of this species are dispersed in the forest due to the abundance of mother trees (*Itô, 2007*; *Itô, 2015*), and it could persist at the seedling stage (*Itô, 2009*). On the other hand, the four species whose colonization probabilities were largest in the gaps are all shade-intolerant (*Shimoda et al., 1994*; *Shibata & Nakashizuka, 1995*). In addition to the four species, *Symplocos prunifolia* is also considered to be a shade-intolerant species though it is an evergreen (*Fujii, 1994*). However, *Symplocos prunifolia* and *Triadica sebifera* are unpalatable plants for sika deer (*Shimoda et al., 1994*; *Hashimoto & Fujiki, 2014*). *Triadica sebifera* is an alien species in Japan. The species is unpalatable for sika deer (*Shimoda et al., 1994*; *Hashimoto & Fujiki, 2014*), and as of 2002, it was increasing on Mt. Kasugayama (*Maesako, Nanami & Kanzaki, 2007*). *Shimoda et al. (1994)* studied the deer effects on pioneer species on Mt. Kasugayama, where deer population density was high; the authors found that pioneer species including *Zanthoxylum ailanthoides* and *Callicarpa mollis* emerged in gaps but rarely survived or matured due to deer foraging pressure.

In the study site, few large stems (height ≥ 50 cm) of *Zanthoxylum ailanthoides*, *Callicarpa mollis*, and *Carpinus tschonoskii* were found, though a greater number of those of *Quercus glauca*, *Cleyera japonica*, and *Eurya japonica* were found in the quadrats that were not affected by the mass oak mortality (*Itô, 2015*). The latter species are evergreen trees or shrubs, and they had been dominant at least since 1992. In addition to these species, large plants of *Symplocos prunifolia* and *Triadica sebifera* were found in the gap quadrats created, expanded, or affected by the mass mortality (*Itô, 2015*), and the high colonization rate of these species in the gaps might contribute to their regeneration. *Suzuki (2013)* pointed out that succession after the abandonment of coppices in which there was gap creation under deer herbivory pressure would increase dominance by shade-tolerant and herbivory-tolerant species. In the present study, the two shade-intolerant and deer-unpalatable species frequently colonized into the gap quadrats and grew up. The difference may be due to the abundance of the two species or overall differences in species composition; for example, the shade-tolerant and herbivory-tolerant species *Maesa japonica* (Thunb.) Moritzi et Zoll. present in the gap plots of (*Suzuki, 2013*) was not found in the present study site.

*Horsley, Stout & DeCalesta (2003)* conducted a field experiment in a hardwood forest in Pennsylvania, North America, and showed that fern abundance increased in thinned and clear-cut stands with an increase in deer density while the abundance of *Rubus* and tree seedlings decreased. *Horsley, Stout & DeCalesta (2003)* pointed out that the established ferns could interfere with the buildup of tree seedlings and the restoration of diversity. *Obora, Watanabe & Yokoi (2013)* studied the effect of deer herbivory on forest regeneration after the mass mortality of oak trees and found that only some unpalatable fern species increased under deer herbivory, while many species including trees increased

within exclosures. *Obora, Watanabe & Yokoi (2013)*, however, also mentioned that the tree species were suppressed by shrub and grass species even in the exclosures. This suggests the difficulty of forests regenerating from damage caused by the mass mortality of oak trees. A similar issue was also reported in *Itô, Kinuura & Oku (2011)*, in a forest subjected to mass mortality and whose floor was dominated by dwarf bamboo. In the present study, such shrub or grass species were lacking, but a deer-unpalatable fern, *Hypolepis punctata* (Thunb.) Mett. ex Kuhn, partially colonized the gap quadrats (*Itô, 2015*). Such deer-unpalatable ferns and/or herbaceous species may suppress regeneration of tree species, and colonize gaps, resulting in dominance of the forest floor.

## CONCLUSION

Gaps that were created, expanded, or affected by the mass mortality of oak trees might increase colonization of pioneer species. Shade-intolerant species such as *Callicarpa mollis*, *Zanthoxylum ailanthoides*, *Carpinus tschonoskii*, *Triadica sebifera*, and *Symplocos prunifolia* were estimated to more frequently colonize the gaps. Among them, deer-unpalatable *Symplocos prunifolia* and *Triadica sebifera* may be more likely to survive or mature under foraging pressure of deer, while deer-palatable species such as *Callicarpa mollis* and *Zanthoxylum ailanthoides* may be unlikely to grow under such pressure. This may change the species composition in regenerating stands.

In the future, deer-unpalatable species such as *Symplocos prunifolia* and *Triadica sebifera* may dominate the understory within the gaps that are created, expanded, or affected by the mass mortality of oak trees rather than the current dominant species such as *Eurya japonica* and *Quercus glauca*, while these current dominant species may retain their dominance within unaffected areas owing to their abundance and shade-tolerance under the current magnitude of deer pressure.

## ACKNOWLEDGEMENTS

I thank Dr. K Hirayama (Kyôto Prefectural University) for cooperating with the fieldwork and the Kyôto-Ôsaka District Forest Office for supporting this work. I also thank Dr. H Iijima (Yamanashi Forest Research Institute) for reading a previous version of the manuscript and for comments. Computational calculations were conducted on the high-performance cluster computing system of AFFIT, the Ministry of Agriculture, Forestry and Fisheries, Japan.

### Funding

The author received JSPS KAKENHI Grant Number JP26450215. The funders had no role in study design, data collection and analysis, decision to publish, or preparation of the manuscript.

## Grant Disclosures

The following grant information was disclosed by the author:
JSPS KAKENHI: JP26450215.

## Competing Interests

The author declares that they have no competing interests.

## Author Contributions

- Hiroki Itô conceived and designed the experiments, performed the experiments, analyzed the data, contributed reagents/materials/analysis tools, wrote the paper, prepared figures and/or tables, reviewed drafts of the paper.

## Field Study Permissions

The following information was supplied relating to field study approvals (i.e., approving body and any reference numbers):

The Kyôto-Ôsaka District Forest Office and the Forestry and Forest Products Research Institute agreed to conduct the study cooperatively in the national forest.

## Data Availability

The raw data has been supplied as a Supplementary File.

## Supplemental Information

Supplemental information for this article can be found online at http://dx.doi.org/10.7717/peerj.2816#supplemental-information.

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
