# Peer review of "Changes in understory species occurrence of a secondary broadleaved forest after mass mortality of oak trees under deer foraging pressure"

_PeerJ, doi:10.7717/peerj.2816_

## Round 0.1 · original submission · Major Revisions

While the reviewers see this work as publishable in PeerJ, all three agree that the paper could be substantially more effective with a careful and substantial revision. Most of their suggestions focus on several common themes: a) providing more conceptual background and placing this study more fully in context of prior work, b) focusing discussion on interpretation of results in ecological context (as one reviewer puts it, discussion should "guide" the reader through your findings in light of your research questions), and c) providing some interpretation of results specifically for readers less familiar with Bayesian methods.

Reviewer 1 ·

Basic reporting

My background in Bayesian statistics is weak. This perhaps contributes to my difficulty understanding the manuscript. While the abstract reveals how individual species responded through time (lines 21-27), the results section do not make this clear. Moreover, the Table and Figure legends do not help a reader with a weak grasp of Bayesian statistics understand the data. At a minimum, it would be useful to know what a negative or positive posterior probability corresponds to. In Fig 1, for example, do negative values indicate a lower probability of occurrence in 1992? How do I interpret the X-axis? That is, how would I interpret a -3 versus a -4?

The research is not placed in a broader framework of knowledge, as the discussion is quite narrowly-focused.

Experimental design

The author’s objective is to estimate the specific probabilities of survival and colonization in forest understory species following mass morality of oak in a permanent 0.5 ha plot in the Ginkakuzi-san National Forest. A Bayesian framework is appropriate for this question.

Validity of the findings

I am unable to judge.

Additional comments

Line 57 – Deer browsing can also fail to inhibit browse-tolerant species, not just unpalatable species

Line 60 – What is meant by “damaged forests?” Please define.

Line 82 – the term “earth outflow” is not clear. Does this mean slope stabilization, flood control, water regulation?

Line 117 – How were gaps defined operationally? In other words, how does one draw a line between inside and outside of a gap?

Reviewer 2 ·

Basic reporting

The article is very well written. However, I suggest that additional details be added for background information (e.g, changes in deer populations over time) and to bolster the discussion and conclusions, which are currently scant. I also suggest including some basic summary statistics of the survey data to help readers unfamiliar with Bayesian statistics, even if they are somewhat similar to the previous publication of these data.

Experimental design

The goal of the research is clearly defined but, without reading the previous publication, I am skeptical that these new analyses add much to the story that was already reported. If there is substantial new information presented here, it needs to be much more thoroughly differentiated from the earlier paper.

A map showing the spatial relationships of the gaps would be helpful. Given that only one site was used, if the gaps are fairly close together, they may not be independent of one another.

I know little about Bayesian statistics so I apologize for not being able to comment on this portion of the methods.

Validity of the findings

As stated above, I am concerned that only a fairly small area of a single site was used for this study and that samples were not independent. However, I recognize that Bayesian statistics may help overcome some of these problems but I am not familiar enough with these analyses to properly evaluate them. Perhaps adding some details about your analyses for other readers like me would allay some concerns.

Additional comments

My primary advice is to better differentiate this paper from the previous publication and add more details throughout.

·

Basic reporting

I have made some comments in the pdf about how to improve presentation. The results need to be rewritten to guide the reader through the findings by explaining what the summary stats mean for a given species. As written, the results are a lot of summary values without any sort of context.

Experimental design

While an area larger than 0.5 ha or more plots would be ideal, I think that the repeated measurements going back over 20 years make this study valuable.

Validity of the findings

I think the findings are valid, but need to be better presented.

Additional comments

I have provided comments within the provide pdf. To summarize my recommendations:
1. Provide some information (if available) in the introduction that describes the pre-mortality gap dynamics of these forests.
2. Present your results in a way that not only presents summary values, but guides the reader through the results so s/he understands how the presence/survivorship have changed.
3. Much of the discussion belongs in the results. Move the highlighted sections and focus on the larger findings of the study in the discussion. Emphasize how these forests will look in the future following the combined impacts of deer and mortality.

---

## Round 0.2 · Minor Revisions

I apologize for the delay in handling of your revision. I had hoped to find a reviewer who might focus specifically on your statistical approaches, but have been unable to do so. Since I have a review from one of the original reviewers, I decided to serve as a second reviewer myself. My comments are included in an attached pdf file

In my judgment, the paper is appropriate and acceptable for PeerJ, but there are enough minor edits called for that I will ask you to address them before a final acceptance. None of the suggested changes concern the substance or arguments of your paper; all of them concern language and clarity of presentation. I think it will be a very simple task to address them (and, if any of them seem to change your meaning, you are free to reject them).

I anticipate quick acceptance following this minor revision.

Reviewer 1 ·

Basic reporting

No comments

Experimental design

No comments

Validity of the findings

No comments.

Additional comments

Lines 360 - 361. Fencing that keeps deer away from vegetation are termed exclosures, whereas enclosures usually refers to fences that keep deer in a plot. Double check the meaning of these sentences.

Line 379. Consider reading and citing Horsley et al. (2003) here. They did an excellent study with controlled deer densities over several years, and showed a causal relationship between deer density, deer browsing, and shifts in future forest composition through altered recruitment patterns. This seems highly relevant to your own findings, and puts your work in a more global context.

Horsley et al. (2003). DOI: 10.1890/1051-0761(2003)013[0098:WTDIOT]2.0.CO;2

---

## Round 0.3 · accepted · Accept

Thank you for your work in revising this manuscript